# Enhancing Growth Performance, Organ Development, Meat Quality, and Bone Mineralisation of Broiler Chickens through Multi-Enzyme Super-Dosing in Reduced Energy Diets

**DOI:** 10.3390/ani11102791

**Published:** 2021-09-24

**Authors:** Jacoba I. Bromfield, Louwrens C. Hoffman, Darwin Horyanto, Elham A. Soumeh

**Affiliations:** 1School of Agriculture and Food Science, Gatton Campus, University of Queensland, Gatton, QLD 4343, Australia; j.madiganstretton@uq.edu.au (J.I.B.); darwinhoryanto@hotmail.com (D.H.); 2Bioproton Pty Ltd., Acacia Ridge, Brisbane, QLD 4110, Australia; 3Queensland Alliance for Agriculture and Food Innovation, University of Queensland, Brisbane, QLD 4343, Australia; louwrens.hoffman@uq.edu.au; 4Department of Animal Sciences, Stellenbosch University, Stellenbosch 7906, South Africa

**Keywords:** Natuzyme, production performance, carcass characteristics, bone minerals, energy reduction

## Abstract

**Simple Summary:**

The global population is expected to rise from 7.2 billion as of 2019 to 9.7 billion in 2050, putting pressure on farmers to increase production capacity to ensure food security whilst simultaneously improving food sustainability. Poultry is an important meat, as chickens have high feed efficiencies and short production cycles, making it an affordable, nutritious source of protein. Strategies to improve the production performance of broilers will require significant research; one nutritional strategy is improving the efficiency of feed utilization via the addition of exogenous enzymes into diets. This study aimed to identify the optimal multienzyme, Natuzyme, dose rate at three energy levels based on production performance, organ development, meat quality, and bone mineralization in broiler chickens. Results revealed that all dose rates of Natuzyme were able to mitigate the negative effect of energy reductions. Organ development and meat quality remained consistent across treatment groups, except for the gizzard and meat moisture content, which were affected by super-dosing Natuzyme. Bone mineralization was restored with the inclusion of Natuzyme. In conclusion, super-dosing Natuzyme in reduced energy diets at a dose rate of 700 g/t can improve performance parameters and thus profitability for producers and can improve the sustainability of production.

**Abstract:**

This study identified the optimal multi-enzyme dose rate at three energy levels based on the production performance of broiler chickens. A 42-day grow out trial was conducted using 576 day-old mixed-sex ROSS308 broiler chickens in a 3 × 4 factorial arrangement in a completely randomized design. Diets consisting of three metabolizable energy (ME) levels: standard energy (STD), 150 kcal/kg energy reduction (STD-150), and 200 kcal/kg energy reduction (STD-200), were cross factored with four multi-enzyme inclusion levels (0, 350, 700, and 1000 g/ton). The average daily feed intake and feed conversion ratio increased linearly (*p* < 0.001) as the dietary ME was reduced, and the multi-enzyme addition improved the feed conversion ratio (*p* < 0.05) and mitigated the negative effect of the reduced energy diets (RED) on feed intake and feed conversion ratios. Carcass composition, organ weights, and meat quality were not affected by the experimental diets. The RED decreased abdominal fat weight (*p* < 0.05). Total ash, calcium, and phosphorous contents of the tibia bone were improved (*p* < 0.04) when the RED were supplemented with the multi-enzyme. Super-dosing multi-enzymes in RED mitigates the negative effect of ME reduction on growth performance while maintaining organ development and meat quality and improving bone mineral content.

## 1. Introduction

Food security and sustainability is of great global concern, as the world population is predicted to reach 9.7 billion by 2050 compared to the current 7.2 billion people [1]. Poultry meat is a particularly important protein, as broiler chickens have a high feed efficiency and short production cycle compared to the other livestock, making it an affordable and nutritious source of animal protein. The global poultry meat consumption per capita has risen 16% from 2008–2017 and is expected to increase by an additional 5.5% in the coming decade [2]. To meet the demand, it is essential to improve the sustainability of poultry production systems and the efficiency of feed utilization. The addition of exogenous enzymes to the feed may reduce diet costs without a reduction in growth performance. Identifying the effective enzyme inclusion rate on key production outputs is vital to ensure that producers achieve the best performance and return on their investment.

After the widespread adoption of phytase in commercial poultry diets, a plethora of companies began experimenting with combining multiple enzymes in an attempt to create a synergistic effect. It is accepted that each enzyme in the multienzyme formulation targets a particular compound in the feed, where it helps to break down more macromolecules; however, there is still some debate surrounding this hypothesis [3]. Poultry can naturally produce several of these enzymes; however, the digestive process with endogenous enzymes alone still leaves up to 25% of feed undigested due to the presence of anti-nutritional factors in the feed itself [3]. Previous studies have reported that the addition of a multi-enzyme increases body weight and improves the feed conversion ratio (FCR) due to an improved nutrient utilization [4,5]. A significant volume of literature has been generated in this area, and although results vary, they commonly support this hypothesis [4,6,7,8,9]. The implementation of enzymes into poultry diets can redefine matrix values, as the enzyme alleviates nutrients that were previously unavailable to the bird. This means that nutrients level, including energy, phosphorus, calcium, and protein, can be reduced in the diet without impairing growth performance. Multi-enzyme super-dosing using the enzyme cocktail in a greater dose rate (almost 3 times greater than the commercial dose) has recently been introduced in the animal feed industry to further amplify the effects of the multi-enzymes in the releasing nutrients. This may result in reducing dietary costs by enabling greater nutrient reductions in feed formulation via defining the matrix values for the multi-enzyme super-dose levels.

The addition of multi-enzymes into broiler diets improves the bioavailability and digestibility of the diet, which can increase carcass weight due to improved nutrient utilization [10]; however, there is limited literature regarding the effects of reduced energy diets with multi-enzyme supplementation on meat quality.

Meat quality is a highly complex yet extremely important factor for the producers and consumers of poultry meat [11]. To encourage the additional growth in this area, it is important for producers and distributors to focus on all of the factors that contribute to consumer preferences, notably cost, yield, and quality, which can potentially be addressed by the addition of multi-enzymes.

Therefore, this study aims to redefine current multi-enzyme matrix values by investigating the effects of multi-enzyme super-dosing in reduced energy diets on production performance, organ development, meat quality, and bone mineralization. Sex-specific response was not an objective of this trial, as the main focus was the overall farm performance of the broilers.

## 2. Materials and Methods

All of the experimental procedures were approved by the Production and Companion Animal Ethics Committee of the University of Queensland prior to the commencement of the trial (number: SAFS/510/18/BIOPROTON).

### 2.1. Animals and Diets

A total of 576 1-day-old mixed-sex broiler chickens (ROSS308) were purchased from a commercial hatchery (Woodlands Hatchery, Queensland, Australia) and were transferred to the Queensland Animal Science Precinct (QASP) facility at Gatton Campus, University of Queensland. The birds were individually weighed and randomly assigned to one of twelve experimental groups in a 3 × 4 factorial arrangement in a completely randomized design, i.e., three dietary energy levels and four dietary inclusion levels of multi-enzyme. Each experimental diet was fed to six replicate pens with eight birds in each (*n* = 48 per experimental group). The experimental diets (Table 1) included a standard wheat–corn–soybean diet with three metabolizable energy (ME) levels (standard/no reduction (STD), −150 kcal/kg reduction in ME (STD-150 kcal/kg), and −200 kcal/kg reduction in ME (STD-200 kcal/kg)) and four enzyme inclusion levels (0, 350, 700, and 1000 g/ton). For the experimental diets, three basal diets (with different energy levels) for each growth phase were mixed and divided into four equal portions. To each basal diet portion, a different multienzyme inclusion level (four inclusion levels) was added, and the diet was thoroughly mixed. All other nutrients were supplied to meet nutrient recommendations as specified in the ROSS308 guidelines (Table 1). The multi-enzyme, Natuzyme (Bioproton Pty. Ltd., Acacia Ridge, Queensland, Australia) is a blend that includes phytase (enzyme activity: 1500 u/g), xylanase (enzyme activity: 10,000 u/g), cellulase (enzyme activity: 6000 u/g), amylase (enzyme activity: 400 u/g), protease (enzyme activity: 700 u/g), beta-glucanase (enzyme activity: 700 u/g), and mannanase (enzyme activity: 400 u/g) and is currently recommended at 350 g/t by the manufacturer. The expected and actual phytase activity was tested for a pooled sample of experimental diets with 350 g/ton, a pooled sample of experimental diets with 700 g/ton, and a pooled sample of experimental diets with 1000 g/ton multi-enzyme inclusion rates (Table 2) following molybdate assay previously described [12]. According to the standard procedure, the 42-day grow-out period was divided into three phases (starter diet: day 1–14; grower diet: day 14–28; finisher diet: day 28–42), and the nutrient levels were adjusted accordingly. The birds had ad libitum access to feed and water for the entire trial period. The lighting program, temperature, and humidity followed the ROSS308 guidelines. The lighting program provided 23 h of light at a 30–40 lux intensity and 1 h of dark (less than 0.4 lux) for the first 7 days and a minimum of 4 h darkness and a light period of 10 lux intensity after 7 days. Temperature was set at 32 °C and 40% relative humidity for the first 7 days and a 2 °C reduction per week after 7 days until the temperature reached 24 °C at 27 days and 40% relative humidity. This temperature and relative humidity were maintained until the end of the trial.

Proximate analyses of the experimental diets were conducted for dry matter (DM), total nitrogen (N), ash, crude fat, crude fiber, calcium (Ca), and phosphorous (P) contents following the Association of Official Agricultural Chemists (AOAC) (2005). Dry matter content was determined by drying the sub-samples in an oven at 105 °C for 24 h. The feed samples were analysed for total N using a LECO CNS928 carbon/nitrogen combustion analyser 1.0 (Leco, St. Joseph, MI, USA) following the instructions of the manufacturer, and the crude protein (CP) was calculated using a factor of 6.25 × N. The ash (%) of the feed samples was determined after combustion at 500 °C for three hours in a muffle furnace. The crude fat of the experimental diets was extracted in a Soxhlet apparatus with ether. The residue of the feed samples after ether extraction were dried in a conventional oven at 60 °C to a constant mass (~12 h), and the fat content was calculated as the difference in dry mass before and after ether extraction. Calcium and phosphorous contents were determined by the spectrophotometry combustion method. Each of the feed ash samples were mixed with 5 ml of 6 M hydrochloric acid (HCl) and 35 mL of distilled water, with the solution being filtered into a 250 mL glass bottle and made up to a final volume of 50 mL with distilled water. Thereafter, the Ca and P levels were measured using a Thermo iCAP 6000 Series Inductive Coupled Plasma (ICP) Spectrophotometer (Thermo Electron Corporation, Str. Rivoltana, Rodana, Milan, Italy).

Crude fiber was determined using the Ankom Filter Bag Technique following American Oil Chemists’ Society (AOCS) Method Ba 6a-05. An empty filter bag (F57, Ankom Technology, New York, NY, USA) was weighed. The dried feed samples (1.0 g) were ground and placed into the weighed bag, and the weight thereof was recorded. Each filter bag was heat-sealed and pre-extracted in a beaker with ether. Up to 24 pre-extracted samples were then placed in a bag suspender and were inserted into the fiber analyzer for processing. The instrument then automatically performed all of the necessary procedural steps to digest the samples and rinse them. The samples were then ashed, and the instrument reported on the basis of the organic matter of the samples. Crude fiber is the loss upon the ignition of the dried residue remaining after the digestion of the sample in 0.255 N H_2_SO_4_ and 0.313 N NaOH solutions under specific conditions.

### 2.2. Growth Performance and Sample Collection

Body weight (BW) and feed intake (FI) were recorded weekly, and the average daily gain (ADG), average daily feed intake (ADFI), and feed conversion ratio (FCR) were calculated for the starter, grower, and finisher phases as well as for the total grow-out period. The mortality rate was recorded daily and was used to calculate the mortality corrected FCR.

At the end of the experiment, one bird with a BW similar to the mean BW of the pen was selected and was euthanized by cervical dislocation, and the carcass was soaked in a scalding tank (60 °C water bath) for 2 min, de-feathered using a commercial stainless steel feather plucker (550 mm TUB), and manually eviscerated. During the evisceration process, the birds were dissected, and the organs, including the heart, liver, proventriculus, gizzard, spleen, bursa Fabricius, pancreas, and abdominal fat pad, were collected and weighed. The breast meat was anatomized, and both pectoralis muscles were weighed and used for meat quality analyses. The meat was removed from the right tibia bone before freezing the bone samples for mineral analyses.

### 2.3. Meat Quality Analysis

Post-processing, the chickens were hung in the refrigerator at 4 °C for approximately 18 h. All meat quality analyses, as explained in detail in the following sections, were conducted after the refrigeration period.

#### 2.3.1. Breast Meat pH

The entire breast muscle was removed and weighed for each bird. The pH meat meter (Hanna, HI 98163, Keysborough, VIC, Australia) was calibrated using a of pH 4 and 7, according to manufacturer’s instructions, and was recalibrated after every 6 readings. The pH probe (containing a temperature compensatory point) was placed 1 cm deep into the centre of the breast tissue, and readings were recorded for all 72 samples. The pH probe was cleaned between each sample according to the manufacturer’s guidelines.

#### 2.3.2. Colourimetry

The left-side of the breast (half of the breast) from each chicken was used to record the meat colour using a Chromameter CR-400/410 (Thermo Fisher Scientific Pty Ltd., Waltham, MA USA) set at d:0° (diffuse illumination/0° viewing angle; specular component included), with a standard observer angle of CIE: 2°. The chromameter was placed on a white tile to calibrate it, as per the supplier’s instructions. After a 40-min blooming period, the meter was then placed on the chicken breast meat at optical infinity (minimum thickness of breast meat 15 mm), and the colour was measured at three different sites on the surface of the meat. The reading values of L*, a*, and b* were recorded and repeated for all 72 samples. To calculate the hue and chroma to determine the precise meat colour, the following equations were used [13]:(1)$$\mathrm{Chroma}\;\left(C*\right)=\sqrt{\;{\left(a*\right)}^{2}\;+{\left(b*\right)}^{2}}$$
(2)$$\mathrm{Hue}\;\mathrm{angle}\;\left(h^{\mathrm{ab}}\right)=\tan-1\;\left(\;\frac{b*}{a*}\;\right)$$

#### 2.3.3. Water Holding Capacity

After trimming off (>2 mm thick) the outer surface layer (as drying out may have occurred), precisely 1.00 g of breast meat was cut and placed on a qualitative 2.5 µm 90 mm diameter filter paper. This was placed on an analytical scale with 0.05 g accuracy. An additional piece of filter paper was placed on top of the sample and was labelled. The sample was then pressed between Perspex plates (standard pressure: 588 N) for exactly 30 s. The filter paper was left to dry for approximately 10 min to improve image clarity before taking an image of the filter paper showing the expelled liquid and pressed meat area, with a ruler next to each sample to set a scale. The drip area was then determined using the software ImageJ. This procedure was repeated for all 72 samples, and the water holding capacity calculated as [13]:(3)$$Water\;Holding\;Capacity=total\;water-loose\;water\;\%\;Loose\;Water=\left(b-a\right)\times \frac{0.0084}{1}\times 100\%\;b:\;area\;enclosed\;by\;the\;outer\;front\;\left({\mathrm{cm}}^{2}\right)\;a:\;area\;enclosed\;by\;the\;inner\;front\;\left({\mathrm{cm}}^{2}\right)\;$$

#### 2.3.4. Cooking Loss

A slice of chicken breast meat was cut from the center of the breast, and its weight was recorded. The breast meat was placed in a thin-walled plastic bag and was sealed before being submerged in an 80 °C water bath for 40 min to cook the sample through. After cooking, the sample was chilled, the water was removed from the bag, and the sample was gently blotted dry with absorbent paper and was weighed. Cooking loss was calculated as:% Cooking Loss = [(weight before − weight after) / (weight before)] × 100(4)

#### 2.3.5. Shear Force

Cooked breast meat (refer to “cooking loss” methodology) was refrigerated until it was completely cool (4 °C). The tenderness of the muscles was determined by measuring the Warner–Bratzler (WBS) shear force values (Instron 5543 model, 15 Stud Road Baywater, Melbourne, Victoria, Australia) with a Warner–Bratzler blade (1 mm thick with a triangular opening, 13 mm at the widest point and 15 mm high, 45 mm long, 60° cutting angle). Cooked muscle samples with a 1 cm^2^ surface area were randomly removed parallel to the longitudinal axis of each muscle. The maximum shear force values (N) at a cross head speed of 33.3 mm/s were measured and were recorded for each of the 72 samples.

### 2.4. Meat Chemical Composition

Post-processing, a portion of the raw left breast (between 90–200 g) was removed, and the weight was recorded. Each sample was individually stored in a small sealable plastic bag. The samples were stored at −20 °C until they were freeze-dried. All of the frozen samples were cut into small pieces and were freeze-dried to constant mass (96 h). Water content was calculated from the difference between the fresh mass and the dry mass. Fat extraction of the dried samples was performed in a Soxhlet apparatus with ether. The defatted samples were dried in a conventional oven at 60 °C to constant mass (~12 h), and the fat mass was calculated as the difference in dry mass before and after ether extraction [14]. The protein content of the meat was analyzed by measuring its nitrogen content from the remaining defatted and dried sample using a LECO CN928 carbon/nitrogen combustion analyzer. The combustion temperature was 1100 °C, and approximately 0.3 g of sample was used for nitrogen analysis. The crude protein content was then calculated as N × 6.25.

### 2.5. Bone Mineralization

The right tibia bones of the 72 chickens were defrosted and broken into small pieces using pliers to enable faster drying. The method used for determining bone mineral composition followed the method described by Hsiao and co-workers [15]; however, the defatting of the sample was not performed due to minor differences reported in the bone mineral composition in the tibia samples, which was able to be determined with or without lipid extraction [16]. In addition, the lipid extraction procedure is a rate-limiting step, and the use of chemical solvents has environmental concerns [16]. The entire tibia bone was collected from the right leg of each broiler chicken at 42 days and was any adhering tissue was cleaned off. The bones were dried to a constant weight at 105 °C and were then ashed in a muffle furnace at 600 °C. The tibia ash was finely ground and dissolved in concentrated HCl for mineral determination (5 mL of 6 M hydrochloric acid and 35 mL of distilled water), and the solution was filtered into a 50 mL glass bottle and was made up to a final volume of 50 mL with distilled water. Thereafter, the Ca and P levels were measured using a Thermo iCAP 6000 Series Inductive Coupled Plasma (ICP) Spectrophotometer (Thermo Electron Corporation, Str. Rivoltana, Rodana, Milan, Italy). The Ca and P composition (g/kg in ash) was calculated using the iTEVA Analyst software (Version 22.0.51).

### 2.6. Statistical Analysis

Analysis of variance was performed on the least square means (LSM) values of the pens using mixed models and two-way ANOVA of Statistical Analysis System (SAS) [17], with the ME levels and multi-enzyme dose rates as the main effects. Orthogonal polynomial contrast coefficients were used to determine linear and quadratic effects of energy levels and multienzyme dose rates if the interactions were significant. All *p* ≤ 0.05 of the values were deemed statistically significant, and all *p* ≤ 0.14 values were considered a tendency. The reported LSM were separated, and differences were evaluated using Tukey’s post hoc test.

## 3. Results

The analysed composition of the experimental diets is reported in Table 1. The analysed composition had discrepancies with the calculated composition.

### 3.1. Growth Performance

The growth performance data, including final body weight (FBW), average daily gain (ADG), average daily feed intake (ADFI), and feed conversion ratio (FCR), for the birds fed with experimental diets are presented in Table 3. The table presents the interactions of multi-enzyme dose rates and reduced-ME diets as well as the main effects of metabolisable energy (ME) levels and the main effects of multi-enzyme dose rates on growth performance parameters. Interactions were noted for the FBW and ADG. Birds fed with STD-150 kcal/kg and no multi-enzyme diet had the lowest FBW (*p* ≤ 0.05) and ADG (*p* ≤ 0.05), which was not different to the STD and no multi-enzyme diets, where all of the other experimental groups had a similar final FBW and ADG. The reduced FBW and ADG were not observed in birds fed the STD-200 kcal/kg diet, which was not expected. The increasing multienzyme dosage linearly improved both the FBW (*p* = 0.04) and ADG (*p* = 0.04) (Table 3). The addition of the multi-enzyme at 350 g/ton to STD-150 kcal/kg, 700 and 1000 g/ton to both reduced ME diets tended to increase ADFI (*p* = 0.08); however, the STD-200 kcal/kg diet had a similar ADFI with no multi-enzyme added. The FCR tended to be the lowest for the STD diet with or without the multi-enzyme addition (*p* = 0.12), which was no different to the STD-150 kcal/kg diet with multi-enzyme added at 350, 700, and 1000 g/ton. The reduced ME diets and the STD-150, and STD-200 kcal/kg diets with no multi-enzyme addition had a tendency for a greater FCR, although the addition of the multi-enzyme at 700 and 1000 g/ton to the STD-200 kcal/kg diet did not improve the FCR.

The dietary ME levels did not influence the FBW and ADG. Dietary ME levels had a significant effect on ADFI (*p* ≤ 0.001) and FCR (*p* ≤ 0.001), where reducing the dietary ME level linearly increased the ADFI to compensate for the energy reduction. This resulted in a linear increase in FCR as the energy was reduced. These results indicate that at all dose rates, the multi-enzyme was able to alleviate the negative effects of the reduced energy diets on ADG, ADFI, and FCR. The addition of the multi-enzyme to the diets (main effects of multi-enzyme dose rates) improved FBW (*p* = 0.04) and ADG (*p* = 0.04) linearly but not quadratically. The multi-enzyme addition had no effects on ADFI; however, a 350 g/ton multi-enzyme dose rate resulted in the lowest FCR (*p* = 0.05), but the 700 and 1000 g/ton multi-enzyme dose rates were not different—neither at the 350 g/ton dose rate nor the STD diet with no multi-enzyme addition.

### 3.2. Organ Development

The effects of the inclusion of a multi-enzyme to reduced-ME diets at different dose rates on carcass composition and organ weight are presented in Table 4. Only the gizzard weights showed an interaction between the main effects. The gizzard weight was significantly different between the experimental groups, and the group with 700 and 1000 g/ton multi-enzyme dose rates added to STD-200 had heavier gizzards, which was similar to the STD-150 kcal/kg diet with no multi-enzyme, the STD-150 kcal/kg diet with 350 g/ton multi-enzyme, and the 1000 g/ton multi-enzyme added to the STD diets (*p* ≤ 0.05); this trend tended to be linearly (*p* = 0.091) correlated to energy levels. No other carcass composition parameters and/or organ weights, including the weights for the carcass, breast, heart, liver, bursa, spleen, pancreas, abdominal fat pad, and proventriculus, were affected by the multi-enzyme dose rates or the interaction.

Reducing dietary ME level, however, tended to increase the breast weight (*p* = 0.059), and significantly reduced abdominal fat weight (*p* = 0.006). Additionally, the heart was significantly bigger in the STD-200 kcal/kg diet compared to the STD diet but was not different compared to the STD-150 kcal/kg diet (*p* = 0.046).

### 3.3. Meat Quality

The effects of multi-enzyme addition to reduced-ME diets on meat quality parameters and proximate analyses are presented in Table 5 and Table 6, respectively. There were no interactions between the main effects for most of the meat quality parameters and chemical composition, except for the breast pH. The breast pH tended to increase when 350 and 700 g/ton multi-enzyme was added to the STD diet and then reduced at the 1000 g/ton multi-enzyme dose rate (*p* = 0.061). A similar increase tendency was observed in breast pH when multi-enzyme was added at 350 and 700 g/ton to the STD-150 kcal/kg diet, and it then decreased when 1000 g/ton multi-enzyme was added to the STD-150 kcal/kg diet. The breast pH had a different response when the multi-enzyme was added to the STD-200 kcal/kg diet, where 350 and 700 g/ton multi-enzyme dose rates tended to decrease breast pH and where 1000 g/ton tended to increase it.

Reducing the dietary ME level (main effects of dietary ME level) resulted in the meat having a significantly lower dry matter content in the STD-150 and STD-200 kcal diets (*p* = 0.019) compared to the standard diet, but there was no difference between the two reduced ME diets (Table 6). In addition, reducing the dietary ME tended to linearly decrease the meat ash content (*p* = 0.131).

The experimental group fed with 350 g/ton multi-enzyme had a greater meat ash content compared to the 700 and 1000 g/ton dose rates, but this was not different to the control diet with no multi-enzyme added (*p* = 0.037). Increasing the multi-enzyme dose rate tended to reduce the breast meat crude protein (*p* = 0.068) and the crude fat (*p* = 0.064) contents (main effects of multi-enzyme dose rates).

### 3.4. Bone Minerals

The bone total ash and mineral contents of the birds when fed with the experimental diets are presented in Table 7. There were significant interactions between the main effects for the total ash and for the phosphorous and calcium levels. The addition of the multi-enzyme at 1000 g/ton to the STD diet reduced the total ash content of the tibia bone (*p* = 0.036) compared to STD no multi-enzyme and STD with 350 g/ton dose rates, but this was not different to the STD with 700 g/ton, STD-150 kcal/kg with no multi-enzyme, and STD-200 kcal/kg with 350 g/ton multi-enzyme dose rates. The STD-200 kcal/kg with 1000 g/ton multi-enzyme dose rates had the greatest total ash content, which was similar to all of the experimental diets, except for the diets with the STD with 1000 g/ton and STD-150 kcal/kg with 350 g/ton multi-enzyme dose rates.

The calcium content of the tibia bone ash was equally higher (*p* = 0.02) at the STD with no multienzyme, STD with 700 g/ton, SRD−150 kcal/kg with 350, 700, and 1000 g/ton, STD-200 kcal/kg with no multi-enzyme, and STD with 1000 g/ton multi-enzyme dose rates.

The STD-150 kcal/kg with no multi-enzyme had the lowest calcium content in the tibia ash, which was not different to the STD ME diets with 350, 700, and 1000 g/ton, STD-150 kcal/kg with 1000 g/ton, and STD-200 kcal/kg without or with multi-enzyme at all dose rates (*p* = 0.02).

The phosphorus content of the tibia ash was the greatest for the STD diet without multi-enzyme (*p* = 0.040), but there was no different when it was compared to the STD with 700 g/ton, STD-150 kcal/kg with 700 and 1000 g/ton, and STD-200 kcal/kg with no multi-enzyme dose rates.

Reducing dietary ME levels tended to linearly increase the total ash content of the tibia (*p* = 0.101) as pertaining to the energy levels. Multi-enzyme dose rates tended to reduce the phosphorous content of tibia ash (*p* = 0.063). The other mineral contents of the tibia bone, including magnesium, potassium, sodium, sulphur, iron, and zinc, did not differ, neither with the dietary ME reduction, nor with the multi-enzyme inclusion rates.

## 4. Discussion

Feed analyses showed minor discrepancies to the calculated nutrient levels, which may have contributed to deviations in the observed results from our hypothesis. The differences between the analysed composition compared to the calculated composition may be due to the 2019 Australian drought, which resulted in a feed supply shortage, which may have led to poorer-quality cereal grains and soybean meal in the diet.

### 4.1. Growth Performance

The hypothesis for the trial was that multi-enzyme super-dosing would compensate for the reduced-ME content of the diets and that these energy saving effects of the multi-enzyme would restore the growth performance parameters in birds fed with reduced-ME diets. The aim was to re-define matrix values for different multi-enzyme inclusion rates to be used for cost-effective feed formulation practices in the poultry feed industry. Currently, a 70 kcal/kg ME reduction is recommended when 350 g/ton multi-enzyme is supplemented. The results revealed that reducing the dietary ME content did not affect ADG and FBW, which was at the cost of significantly increased ADFI and consequently higher FCR. Super-dosing the multi-enzyme, however, did compensate for reduced ME content and restored the growth performance of broiler chickens, including ADFI and FCR. This fact was evidently reflected in the FCR value for the STD-150 kcal diet with no multi-enzyme inclusion, which was significantly reduced from 1.60 to 1.48 with the 350 and 700 g/ton multi-enzyme inclusion rates. Adding the multi-enzyme at 1000 g/ton did not result in a further improvement to the FCR even though it was still significantly better than STD-150 kcal/kg with no multi-enzyme (FCR 1.51 vs. 1.60 for STD-150 kcal/kg plus 1000 g/ton multi-enzyme and STD-150 kcal/kg with no multi-enzyme). Only birds consuming the STD-150 kcal/kg with no enzyme included had a significantly lower final body weight and ADG. It was expected that the birds on the STD-200 kcal/kg and no enzyme diet would have a worse growth performance than birds fed the STD-150 kcal/kg diets; however, the opposite results were observed, and the birds on STD-200 kcal/kg outperformed the birds on the STD-150 kcal/kg diet. The birds on the STD-200 kcal/kg diet had the highest ADFI, which resulted in the highest daily ME intake, which might be the reason for a heavier final BW and higher ADG in this group compared to the birds on the STD-150 kcal/kg diet.

These production performance results are predominantly positive and are consistent with the current literature [18,19]. A lack of significant difference in FCR reveals that multi-enzyme inclusion mitigated the negative effect of reduced ME on FCR, highlighting the energy saving effects of the multi-enzyme. As presented in Table 3, the birds on a reduced-ME diet and no multi-enzyme inclusion compensated for the energy deficiency by maintaining higher feed consumption (*p* ≤ 0.001). The higher feed intake compensates for the reduced dietary ME level, as the total ME intake was increased, and neither final BW, nor ADG were therefore affected by the reduced-ME diets when the multi-enzyme cocktail was added at all levels. Therefore, although there is no difference in ADG between the standard and reduced-ME diets due to the increased feed intake, there is a significant increase in ADFI to compensate for the reduction in energy, which results in differences in FCR between the STD and reduced-ME diets.

The results of the trial demonstrated that the addition of the multi-enzyme cocktail into reduced-ME diets did not affect ADFI, which was consistent with previous reports where multi-enzymes were added to corn–soybean diets [19]. The birds fed on a negative control diet with reduced nutrient density had poorer weight gain and feed efficiency than those given the positive control diet with the recommended nutrient density. Supplementing the multi-enzyme cocktail to both negative and positive control diets improved the weight gain and feed efficiency compared to the no multi-enzyme inclusion. In the aforementioned study, similar to our findings, there was no effect of multi-enzyme inclusion on feed intake either.

Multi-enzyme super-dosing at 1000 g/ton significantly improved FBW and ADG compared to the control diet (main effects of the multi-enzyme inclusion rate). The overall improvement in performance when the multi-enzyme was added to the diet is due to the increased nutrient bioavailability and ability of the enzymes to combat antinutritional factors present in the ingredients. Corn, soybean meal, and wheat all contain antinutritional factors that decrease performance if not remedied. Soybean meal contains trypsin inhibitors, which block the degradation of protein, thus decreasing the overall availability of protein in the diet [20], while corn and wheat contain phytic acid, which limits phosphorus, calcium, and zinc bioavailability [3]. The multi-enzyme cocktail that was evaluated in the present work contains protease, which alleviates the effect of the trypsin inhibitors in soybean meal, as it improves protein hydrolysis in the presence of trypsin and therefore increases the digestible protein content of the diet [21]. Additionally, the multi-enzyme cocktail also contains phytase, which breaks the bond between phytic acid, phosphorus, calcium, and zinc to increase mineral availability [22]. However, there was no difference in the final BW, ADG, or FCR between the different enzyme dose rates, suggesting that increasing the dose rate to “super-dosed” levels is not necessary to improve performance, which is consistent with other findings [5,23].

Overall, all performance traits in this experiment revealed a positive result in terms of ME level and multi-enzyme interaction. Body weight values for all of the experimental groups were similar and statistically greater than the STD-150 kcal/kg diet. This is to be expected, as the other experimental groups were either a STD diet or diets that had their energy reduction compensated for by the addition of the multi-enzyme cocktail, with the exception of the STD-200 kcal/kg diet. As ADG is a factor of final BW, the observed trend was similar for ADG. However, a point of interest is the better growth performance of the birds on the STD-200 kcal/kg diet compared to the STD-150 kcal/kg diet. This could be attributed to the fact that the STD-200 kcal/kg diet could have triggered the compensatory growth mechanism in the chickens. Compensatory growth occurs when an animal has restricted access to feed or nutrients and consequently increases feed consumption, utilization, and conversion efficiency. Although the STD-150 kcal/kg diet also had an energy reduction, this reduction may not have been sufficient to trigger the metabolic response of compensatory growth. Compensatory growth from feed, energy, and protein restriction has been documented in poultry, with Sunder et al. [24] and Leeson [25] observing similar findings. Although compensatory growth enables the birds consuming reduced energy diets to perform similarly to the birds fed a standard diet in terms of FCR, there was a large increase in mortality rate (Table 3). This indicates that producers cannot improve flock performance solely through manipulating compensatory growth, as the flock mortality rate will be greater. The ROSS308 guidelines (2014) indicate that expected mortality rate in a flock is 5%, whereas reduced energy with no multi-enzyme supplementation yields a greater mortality rate—double that figure. This high incidence of mortality decreased when the multi-enzyme cocktail was added into the reduced energy diets, thus indicating that a multi-enzyme mixture is able to compensate for the reduced ME in the diets.

### 4.2. Organ Development

Organ development is an important factor in poultry production, and it is crucial to minimize fat depositions in the organs to prevent poor carcass and meat quality. In the present trial, the significantly higher abdominal fat deposition in the STD diet compared to the reduced-ME diets was to be expected, as the ME intake and ME:Protein intake was greater for birds fed with the STD diet. In contrast, the birds on the reduced-ME diets needed to devote and utilize the limited energy supply to maintain functions, thus less energy was left for abdominal fat deposition. Reduced abdominal fat is an advantage in poultry production, as consumers do not desire chicken carcasses with a high fat content [26].

The ME levels and multi-enzyme dose rate interaction had no significant effects on organ weights, except for gizzard weight. There was a tendency for the gizzard to be heavier in the treatment groups with reduced-ME levels and multi-enzyme supplementation, particularly in the STD-200 kcal/kg diet (Table 4). This interaction was mainly correlated to the effect of reduced energy diets that tended to linearly increase the gizzard weight (*p* = 0.09). This may be due to the increased muscularity required by the gizzard to break down drier feed, as the energy reduction was a result of the decrease in oil quantity in the diet. Dry feed is generally retained in the gizzard–proventriculus system longer than moist feed, as previously reported [27], thus resulting in an increase in the gizzard muscle mass.

### 4.3. Meat Quality

Breast pH tended to be influenced by the multi-enzyme addition to the reduced-ME diets (*p* = 0.06), which was led by energy levels when tested as a main effect (Table 5). The results revealed that as the energy level was reduced in the diet, the breast pH became more acidic. Post-mortem, rigor mortis ensues, whereby muscle metabolism changes from aerobic to anaerobic and where lactic acid is produced as a by-product, causing a decline in pH [28]. Although rigor mortis in all biological organisms eventually leads to the production of lactic acid and thus a decline in pH, the energy source of the diet and quantity in the muscle ante mortem may influence the ultimate pH. As indicated by production performance, the feed intake increased in reduced energy diets to compensate for the lack of energy. To create an energy reduction, dietary soybean oil was replaced with corn, which is rich in carbohydrates, and the glucose is stored in the muscle as glycogen. It is a simpler process for the body to utilize glycogen and to convert this to energy via the anaerobic pathway with lactic acid as by-product rather than by utilizing the fat; thus, it can be speculated that the lower pH in the reduced energy diets may be due to the increased carbohydrate content of the diet, resulting in higher glycogen storage within the muscles.

In the present study, the values relating to the meat lightness (L*) were above 57 (Table 5), indicating meat that was lighter than normal; however, there was no significant difference among the experimental groups. It is likely that the lighter breast might be related to the temperature of the scalding bath used for de-feathering. The scalding bath was set at 60 °C, which might have been too hot, causing the chicken breast to be slightly poached, resulting in slightly higher lightness values. It has been reported that breast colour lightness (L*) values above 49 are suggestive of a poor WHC and increased shear force and a low pH, as the lighter colour is indicative of increased reflectance caused by the increased water leakage that accumulates on the surface of the breast and reflects the light rays during the measurement of the colour [29]. In the current trial, no differences were observed for the breast lightness and the WHC among the experimental groups; however, all of the recorded values were greater than the values reported in the literature for both normal and pale meat [30]. The result of the current study is consistent with the data of other researchers who have reported a lighter meat colour (greater L* value) when broiler chickens were fed on low-energy diets [31,32]. Reports [32] also showed that a low-ME enzyme-added diet increased the yellowness of the breast muscle (a*) in comparison with the standard ME and low-ME diet with no enzyme addition, which is slightly different to our findings. Lightness (L*) and redness (b*) were also not influenced by the dietary treatments in their study.

A higher WHC relates to an acidic pH. As the pH of poultry meat decreases to 5.3–5.4, it reaches the isoelectric point for many major proteins [33]. Proteins with a charge that is closer to neutral imply that there is little polarity in the molecule; therefore, water is not attracted to it and will thus purge itself from the meat. A less acidic pH is associated with greater protein polarity, allowing an increased WHC, which is reflected in the present results (Table 5).

Birds on the reduced energy diets tended to have a heavier breast weight (Table 4; main effects of the energy levels). This might be due to a slightly greater protein intake (gram per day) for birds on the reduced energy diets because of increased ADFI in response to the lower energy content. These findings agree with the results reported [34], where breast muscle weight (%) tended to decrease when the dietary ME levels decreased from 2805 to 2997 kcal/kg. A similar pattern could be observed in the results reported elsewhere [33]; however, the broiler chickens on the low-ME diet with or without enzyme addition had heavier carcasses compared to the birds on the standard ME diet.

The other meat quality parameters were not affected by the experimental diets. These outcomes are in line with the results reported [35], where with the exception of meat hardness, low-ME diets supplemented with enzyme did not affect any other meat quality parameters compared to the standard diet. Similarly, it has been reported that enzyme supplementation does not affect the physical properties of breast meat, the including pH and WHC, in broiler chickens [36]. Our data were also in agreement with the findings from [37], which observed no differences in meat composition or meat quality parameters when the birds were fed with commercial enzymes, including xylanase and phytase, applied individually or in combination.

### 4.4. Bone Mineralisation

Bone mineral contents were altered by the multi-enzyme addition to the reduced-ME diets (Table 7). Our results show that multi-enzyme at 350 and 700 g/ton when added to the STD-150 kcal/kg diet and that 1000 g/ton when added to the STD-200 kcal/kg diet restored the bone calcium content to the level of the STD diet. A similar outcome was observed for the bone phosphorous content at the 350 and 700 g/ton multi-enzyme dose rates when added to the STD-150 kcal/kg diet. Reducing the ME level of the diet tended to increase the total ash of the tibia bone and increasing the multi-enzyme inclusion rates tended to reduce the bone phosphorous content. These results were in line with the findings of [38], where the supplementation with an enzyme complex to the negative control diets restored bone mineralization to the level of the positive control diet in meat ducks. Similarly, an enhanced tibia ash and bone mineralization with a phytase addition to a control diet have been reported [39,40]. The bone mineral concentration responded to the multi-enzyme supplementation, as it contains phytase. It is believed that the tibia ash and mineral concentration may be more sensitive indicators of the mineral utilization efficiency than the growth performance parameters are.

## 5. Conclusions

The addition of a multi-enzyme cocktail to the diets mitigated the negative effects of energy reduction on growth performance parameters. This highlights the cost-saving effects of multi-enzymes, as expensive ingredients such as oil can be reduced from the diet without any negative influence on production performance, meat quality, and organ development if supplemented with the specific multi-enzyme cocktail that was evaluated. In conclusion, based on the outcomes of this study and considering all of the studied parameters, the evaluated multi-enzyme cocktail at a dose rate of 700 g/t could be used in feed formulation practices, saving 150 kcal of metabolizing energy. This can replace the current matrix value recommendation of 70 kcal/kg energy savings when using a commercial dose rate (350 g/t) of multi-enzyme.

## Figures and Tables

**Table 1 animals-11-02791-t001:** Ingredient composition as well as the calculated and analysed nutrient composition of the experimental diets ^1^.

Diet Composition (g/kg)	Starter (1–14 d)			Grower (15–28 d)			Finisher (29–42 d)		
	STD Energy	STD-150 Kcal/kg	STD-200 kcal/kg	STD Energy	STD-150 Kcal/kg	STD-200 kcal/kg	STD Energy	STD-150 Kcal/kg	STD-200 kcal/kg
Ingredient Composition									
Wheat	400.0	400.0	400.0	400.0	400.0	400.0	450.0	450.0	450.0
Corn	204.7	239.3	250.1	200.0	237.2	249.7	210.2	247.4	259.8
Soybean meal	298.3	293.9	293.0	293.6	286.9	284.7	238.1	231.3	229.6
Soybean oil	42.7	12.5	2.6	58.9	28.3	18.1	58.4	27.7	17.5
L-Lysine HCL	5.0	5.0	5.0	3.5	3.6	3.6	3.4	3.5	3.6
DL-Methionine	4.1	4.0	4.0	3.4	3.3	3.3	3.1	3.1	2.1
L-Threonine	2.6	2.5	2.5	1.8	1.8	1.7	1.6	1.6	1.6
Limestone	15.2	15.3	15.3	13.9	13.9	13.9	12.6	12.7	12.7
Mono-calcium phosphate	17.4	17.3	17.3	15.1	15.0	15.0	13.2	13.1	13.2
Sodium bicarbonate	3.2	3.2	3.3	2.5	2.6	2.6	2.5	2.6	2.6
Salt	1.5	1.5	1.4	2.0	1.9	1.9	1.9	2.0	2.4
Vitamin/Mineral Premix	5.0	5.0	5.0	5.0	5.0	5.0	5.0	5.0	5.0
Coccidiostat	0.5	0.5	0.5	0.5	0.5	0.5	0.0	0.0	0.0
Total	1000.0	1000.0	1000.0	1000.0	1000.0	1000.0	1000.0	1000.0	1000.0
Calculated Nutrients Composition									
ME ^2^ Kcal/kg (MJ/kg)	2850.0 (11.92)	2700.0 (11.30)	2650.0 (11.09)	2950.0 (12.34)	2800.0 (11.72)	2750.0 (11.51)	3000.0 (12.55)	2850.00 (11.92)	2800.0 (11.72)
Crude protein (g/kg)	210.0	210.0	210.0	205.0	205.0	205.0	185.0	185.0	185.0
SID ^3^ Lysine (g/kg)	12.80	12.80	12.80	11.50	11.50	11.50	10.20	10.20	10.20
SID Met + Cys (g/kg)	9.50	9.50	9.50	8.70	8.70	8.70	8.00	8.00	8.00
SID Threonine (g/kg)	8.60	8.60	8.60	7.70	7.70	7.70	6.80	6.80	6.80
Calcium (g/kg)	9.60	9.60	9.60	8.70	8.70	8.70	7.80	7.80	7.80
Avail. Phosphorous (g/kg)	4.80	4.80	4.80	4.35	4.35	4.35	3.90	3.90	3.90
Sodium (g/kg)	1.60	1.60	1.60	1.60	1.60	1.60	1.60	1.60	1.60
Analysed Nutrients Composition									
Dry matter (g/kg)	950.3	951.4	949.6	949.6	949.5	952.3	948.3	952.1	950.4
Crude Protein (g/kg)	207.2	219.3	206.4	196.6	205.4	203.0	201.0	187.2	182.5
Crude fat (g/kg)	67.7	35.6	25.3	85.4	53.0	42.0	85.0	52.1	41.5
Crude fiber (g/kg)	27.5	28.1	28.4	27.2	27.9	27.9	26.7	27.1	27.4
Calcium (g/kg)	14.7	13.8	16.2	11.9	12.0	14.0	10.1	12.5	10.3
Total phosphorous (g/kg)	6.5	5.2	6.6	6.1	6.2	6.6	5.3	5.6	4.7

^1^ Three basal diets with different energy levels (standard/no reduction (STD), −150 kcal/kg reduction in ME (STD-150 kcal/kg), and −200 kcal/kg reduction in ME (STD-200 kcal/kg) were formulated and Natuzyme was added at 0, 350, 700, and 1000 g/ton dose rates into each basal diet for the starter, grower, and finisher phases. Calculated nutrients composition is on “as is” basis, and analyzed nutrient composition is on “DM” basis. ^2^ ME: Metabolisable Energy. ^3^ SID: Standardised Ileal Digestible.

**Table 2 animals-11-02791-t002:** Expected and actual (analysed) phytase activity of experimental diets.

Experimental Diets ^1^	Expected Phytase Activity (u/g)	Actual Phytase Activity (u/g)
350 (g/t)	1.27	1.20
700 (g/t)	2.54	2.51
1000 (g/t)	3.63	3.59

^1^ Three basal diets with different energy levels were formulated, and Natuzyme was added at 0, 350, 700, and 1000 g/ton dose rates into each basal diet for the starter, grower, and finisher phases. The phytase activity (as a reference) is reported for each dosage.

**Table 3 animals-11-02791-t003:** Effects of multi-enzyme super-dosing in reduced energy diets on broiler growth performance parameters.

Interactions		Performance Parameters ^5^, D 1–42				
Energy Level	Multi-Enzyme Dosage	FBW ^1^, g	ADG ^2^, g/day	ADFI ^3^, g/day	FCR ^4^	Mortality%
STD	0	2540 ^a,b^	59.4 ^a,b^	87.1 ^d,e^	1.46 ^e^	0 ^c^
	350 g/t	2564 ^a^	60.0 ^a^	87.5 ^d,e^	1.46 ^e^	4.17 ^a,b,c^
	700 g/t	2543 ^a^	59.5 ^a^	86.5 ^e^	1.46 ^e^	2.08 ^b,c^
	1000 g/t	2622 ^a^	61.4 ^a^	88.8 ^c,d,e^	1.45 ^e^	4.17 ^a,b,c^
STD-150 kcal/kg	0	2396 ^b^	56.0 ^b^	89.2 ^c,d,e^	1.60 ^a^	8.33 ^a,b^
	350 g/t	2649 ^a^	62.0 ^a^	92.0 ^a,b,c^	1.48 ^c,d,e^	6.25 ^a,b,c^
	700 g/t	2633 ^a^	61.6 ^a^	91.0 ^a,b,c,d^	1.48 ^d,e^	8.33 ^a,b^
	1000 g/t	2649 ^a^	62.0 ^a^	93.5 ^a,b^	1.51 ^b,c,d,e^	2.08 ^b,c^
STD-200 kcal/kg	0	2585 ^a^	60.5 ^a^	93.4 ^a,b^	1.55 ^a,b,c^	10.42 ^a^
	350 g/t	2557 ^a^	59.9 ^a^	89.8 ^b,c,d,e^	1.50 ^b,c,d,e^	6.25 ^a,b,c^
	700 g/t	2592 ^a^	60.7 ^a^	94.5 ^a^	1.56 ^a,b^	2.08 ^b,c^
	1000 g/t	2529 ^a,b^	59.2 ^a,b^	90.9 ^a,b,c,d^	1.54 ^a,b,c,d^	4.17 ^a,b,c^
SEM ^5^	-	52.0	1.2	1.8	0.03	2.38
*p*-Value Energy Level × Multi-Enzyme Dose	-	0.051	0.051	0.081	0.118	0.112
Main Effect						
Energy Level	STD	2567	60.1	87.5 ^b^	1.46 ^b^	2.60 ^b^
	STD-150 kcal/kg	2581	60.4	91.4 ^a^	1.52 ^a^	6.25 ^a^
	STD-200 kcal/kg	2565	60.0	92.2 ^a^	1.54 ^a^	5.73 ^a,b^
SEM	-	25.88	0.62	1.31	0.01	1.19
*p*-Value Energy Level	-	0.893	0.896	0 < 001	0 < 001	0.072
Linear	-	0.927	0.935	-	-	-
Quadratic	-	0.643	0.646	-	-	-
Main Effect						
Multi-Enzyme Dose	0	2507 ^b^	58.6 ^b^	89.9	1.54 ^a^	6.25
	350 g/t	2590 ^a,b^	60.6 ^a,b^	89.8	1.48 ^b^	5.56
	700 g/t	2589 ^a,b^	60.6 ^a,b^	90.7	1.50 ^a,b^	4.17
	1000 g/t	2600 ^a^	60.9 ^a^	91.0	1.50 ^b^	3.47
SEM	-	29.88	0.71	1.38	0.02	1.37
*p*-Value Multi-Enzyme Dose	-	0.108	0.110	0.642	0.050	0.470
Linear	-	0.038	0.039	-	-	-
Quadratic	-	0.240	0.243	-	-	-

^1^ FBW: final body weight; ^2^ ADG: average daily gain; ^3^ ADFI: average daily feed intake; ^4^ FCR: feed conversion ratio. ^5^ SEM: Standard Error of Mean; ^5^ Values with different superscripts (a, b, c, d, e) differ (*p* ≤ 0.05).

**Table 4 animals-11-02791-t004:** Effects of multi-enzyme super-dosing in reduced energy diets on broiler chicken organ development.

Interactions		Carcass Composition and Organs Absolute Weight ^1^, g									
Energy Level	Multi-Enzyme Dosage	Carcass	Breast	Heart	Liver	Bursa	Spleen	Gizzard	Pancreas	Abdominal Fat	Proventriculus
STD	0	2017	711	11.1	50.1	5.0	2.7	47.2 ^a,b,c^	5.0	34.7	11.0
	350 g/t	2064	732	12.7	52.3	5.6	2.9	45.3 ^b,c^	4.9	39.2	12.1
	700 g/t	1947	657	12.7	51.9	4.9	2.7	42.0 ^c^	4.7	38.2	10.2
	1000 g/t	2038	698	12.7	53.1	4.7	3.0	46.8 ^a,b,c^	4.5	38.4	11.4
STD-150 kcal/kg	0	1879	666	10.8	50.7	4.6	2.6	46.4 ^a,b,c^	4.5	30.1	10.3
	350 g/t	2088	751	11.7	53.5	4.9	2.5	52.8 ^a,b^	5.1	29.6	10.6
	700 g/t	2065	746	11.1	47.3	4.8	2.4	41.4 ^c^	4.8	28.9	8.7
	1000 g/t	2027	721	12.4	49.9	4.5	2.4	44.9 ^b,c^	4.8	33.1	9.6
STD-200 kcal/kg	0	2031	741	12.7	48.2	4.6	2.5	45.1 ^b,c^	4.9	26.0	9.8
	350 g/t	2107	782	14.5	51.7	5.4	3.1	46.2 ^b,c^	5.1	31.2	12.8
	700 g/t	2052	765	12.7	50.7	5.4	2.5	51.5 ^a,b^	4.7	31.9	10.0
	1000 g/t	2139	774	13.7	49.4	5.5	2.7	54.5 ^a^	5.3	31.7	10.1
SEM	-	87.8	36.2	1.4	3.5	0.6	0.2	3.0	0.4	4.0	0.91
*p*-Value Energy Level × Multi-Enzyme Dose	-	0.794	0.624	0.972	0.965	0.933	0.665	0.049	0.700	0.960	0.811
Main Effect											
Energy Level	STD	2017	699 ^b^	12.3 ^a,b^	51.9	5.1	2.8	45.3	4.8	37.6 ^a^	11.2 ^a^
	STD-150 kcal/kg	2015	721 ^a,b^	11.5 ^b^	50.4	4.7	2.5	46.4	4.8	30.4 ^b^	9.8 ^b^
	STD-200 kcal/kg	2082	757 ^a^	13.4 ^a^	50.0	5.2	2.7	49.4	5.0	30.2 ^b^	10.7 ^a,b^
SEM	-	55.5	21.6	1.0	1.8	0.3	0.1	1.6	0.2	2.5	0.5
*p*-Value Energy Level	-	0.390	0.059	0.046	0.734	0.522	0.157	0.140	0.532	0.006	0.114
TLinear	-	-	-	-	-	-	-	0.091	-	-	-
Quadratic	-	-	-	-	-	-	-	0.234	-	-	-
Main Effect											
Multi-Enzyme Dose	0	1976	706	11.5	49.7	4.7	2.6	46.2	4.8	30.2	10.4 ^a,b^
	350 g/t	2087	755	13.0	52.5	5.3	2.8	48.1	5.0	33.3	11.8 ^a^
	700 g/t	2021	710	12.2	50.0	5.0	2.5	45.0	4.7	33.0	9.6 ^b^
	1000 g/t	2068	731	12.9	50.8	4.9	2.7	18.7	4.9	34.4	10.3 ^b^
SEM	-	59.9	23.7	1.0	2.8	0.3	0.1	1.9	0.2	2.7	0.5
*p*-Value Multi-Enzyme Dose	-	0.318	0.277	0.276	0.753	0.613	0.441	0.394	0.734	0.531	0.034
Linear	-	-	-	-	-	-	-	0.593	-	-	-
Quadratic	-	-	-	-	-	-	-	0.630	-	-	-

^1^ Values with different superscripts (a, b, c) differ (*p* ≤ 0.05).

**Table 5 animals-11-02791-t005:** Effects of multi-enzyme super-dosing in reduced energy diets on meat quality parameters of broiler chicken.

Interactions		Meat Quality Parameters								
Energy Level	Multi-Enzyme Dosage	Breast Weight (g)	Breast pH	WHC ^1^, %	SF ^2^, N	CWL ^3^, %	Breast L* ^4^	Breast a* ^5^	Breast b* ^6^	Hue (◦)
STD	0	711	5.78	71.8	24.1	26.8	59.5	2.2	5.8	60.9
	350 g/t	732	5.81	70.1	20.5	24.6	58.0	2.7	6.5	69.0
	700 g/t	657	5.87	70.9	20.6	25.4	60.2	2.1	6.7	74.6
	1000 g/t	698	5.75	70.8	22.7	26.7	61.3	2.9	7.9	68.9
STD-150 kcal/kg	0	666	5.76	71.0	25.0	22.1	58.1	2.4	5.7	67.5
	350 g/t	751	5.86	71.5	26.1	29.7	57.0	1.7	4.9	64.4
	700 g/t	746	5.87	70.9	21.2	29.5	59.6	2.8	7.4	69.4
	1000 g/t	721	5.80	70.6	22.8	26.1	58.4	2.9	6.2	64.7
STD-200 kcal/kg	0	741	5.80	71.2	23.5	27.0	58.5	1.7	5.9	73.6
	350 g/t	782	5.77	71.3	22.5	26.6	57.1	2.1	6.6	70.1
	700 g/t	765	5.67	72.3	23.6	29.0	59.5	1.9	6.9	71.2
	1000 g/t	774	5.82	72.8	23.0	31.9	58.9	2.2	5.9	68.1
SEM	-	36.2	0.04	0.9	2.5	2.1	1.9	1.0	0.9	6.5
*p*-Value Energy Level × Multi-Enzyme Dose	-	0.62	0.061	0.554	0.822	0.145	0.994	0.906	0.281	0.532
Main Effect										
Energy Level	STD	699	5.81	70.9	22.0	25.9	59.8	2.4	6.7	68.3
	STD-150 kcal/kg	721	5.82	71.0	23.8	26.9	58.3	2.2	6.0	66.5
	STD-200 kcal/kg	757	5.76	71.9	23.1	28.6	58.5	2.0	6.3	70.8
SEM	-	21.6	0.02	0.5	1.5	1.3	1.3	0.9	0.7	4.9
*p*-Value Energy Level	-	0.059	0.186	0.177	0.554	0.169	0.375	0.428	0.370	0.359
Main Effect										
Multi-Enzyme Dose	0	706	5.78	71.3	24.2	25.3	58.7	2.1	5.8	67.3
	350 g/t	755	5.82	70.9	23.0	27.0	57.4	2.0	6.0	67.9
	700 g/t	711	5.80	71.3	21.8	27.9	59.8	2.3	7.0	71.8
	1000 g/t	731	5.79	71.4	22.9	28.2	59.5	2.7	6.6	67.2
SEM	-	23.7	0.03	0.5	1.7	1.4	1.3	0.9	0.7	5.1
*p*-Value Multi-Enzyme Dose	-	0.277	0.782	0.888	0.679	0.316	0.252	0.501	0.097	0.492

^1^ WHC: water holding capacity; ^2^ SF: shear force; ^3^ CWL: cooking water loss; ^4^ L*: lightness; ^5^ a*: redness; ^6^ b*: yellowness.

**Table 6 animals-11-02791-t006:** Effects of multi-enzyme super-dosing in reduced energy diets on meat approximate analyses of broiler chicken.

Interactions		Meat Approximate Analyses ^2^				
Energy Level	Multi-Enzyme Dose Rate	Moisture %	Nitrogen % (DM) ^1^	Crude Protein % (as is)	Ash % (as is)	Crude Fat % (as is)
STD	0	74.7	14.5	19.6	1.3	3.7
	350 g/t	73.9	14.1	19.8	1.4	3.7
	700 g/t	74.8	14.5	19.5	1.3	3.7
	1000 g/t	75.1	14.4	19.2	1.3	3.6
STD-150 kcal/kg	0	74.5	14.7	20.0	1.3	3.8
	350 g/t	75.4	14.5	19.0	1.3	3.6
	700 g/t	75.6	14.6	19.5	1.2	3.6
	1000 g/t	75.2	14.2	19.1	1.3	3.6
STD-200 kcal/kg	0	75.3	14.8	19.4	1.3	3.6
	350 g/t	75.3	14.7	19.3	1.3	3.6
	700 g/t	75.1	14.3	19.1	1.2	3.6
	1000 g/t	75.6	14.1	18.5	1.2	3.5
SEM	-	0.4	0.2	0.4	0.05	0.07
*p*-Value Energy Level × Multi-Enzyme Dose	-	0.173	0.474	0.555	0.280	0.615
Main Effect						
Energy Level	STD	74.6 ^a^	14.4	19.5	1.29	3.6
	STD-150 kcal/kg	75.2 ^b^	14.6	19.3	1.26	3.6
	STD-200 kcal/kg	75.3 ^b^	14.4	19.1	1.24	3.6
SEM	-	0.2	0.1	0.2	0.04	0.05
*p*-Value Energy Level	-	0.019	0.526	0.151	0.131	0.223
Main Effect						
Multi-Enzyme Dose	0	74.8	14.6	19.7	1.27 ^a,b^	3.7
	350 g/t	74.9	14.4	19.4	1.31 ^a^	3.6
	700 g/t	75.2	14.5	19.2	1.23 ^b^	3.6
	1000 g/t	75.3	14.3	18.9	1.25 ^b^	3.5
SEM	-	0.2	0.1	0.3	0.04	0.05
*p*-Value Multi-Enzyme Dose	-	0.363	0.276	0.068	0.037	0.064

^1^ DM: Dry Matter (based on dry matter). ^2^ Values with different superscripts (a, b) differ (*p* ≤ 0.05).

**Table 7 animals-11-02791-t007:** Effects of multi-enzyme super-dosing in reduced energy diets on bone mineral content of broiler chicken.

Interactions		Bone Mineral Content ^1^, g/kg of Tibia Bone Ash								
Energy Level	Multi-Enzyme Dose	Total Ash, g/kg DM	Calcium	Phosphorous	Magnesium	Potassium	Sodium	Sulphur	Iron	Zinc
STD	0	42.3 ^a,b^	376.5 ^a,b^	184.1 ^a^	7.7	4.5	10.6	1.6	237.6	289.8
	350 g/t	43.0 ^a,b^	371.8 ^b,c^	179.2 ^b,c^	7.6	4.4	10.1	1.4	208.2	279.5
	700 g/t	42.1 ^a,b,c^	375.7 ^a,b,c^	181.9 ^a,b^	7.5	4.4	10.1	1.4	223.0	282.8
	1000 g/t	38.5 ^c^	372.9 ^b,c^	180.1 ^b,c^	7.7	5.0	10.6	2.3	201.5	261.0
STD-150 kcal/kg	0	40.1 ^b,c^	369.9 ^c^	179.3 ^b,c^	7.6	5.3	11.4	2.0	243.6	253.9
	350 g/t	43.0 ^a,b^	377.4 ^a,b^	181.4 ^a,b,c^	7.7	4.7	10.7	1.7	239.0	261.5
	700 g/t	43.4 ^a,b^	380.8 ^a^	182.3 ^a,b^	7.3	4.4	10.2	1.4	215.1	275.9
	1000 g/t	42.7 ^a,b^	372.3 ^b,c^	180.0 ^b,c^	7.5	4.9	10.9	1.8	239.9	270.3
STD-200 kcal/kg	0	43.8 ^a^	376.1 ^a,b,c^	182.2 ^a,b^	8.0	4.8	10.8	1.5	244.6	259.0
	350 g/t	40.7 ^a,b,c^	371.0 ^b,c^	180.4 ^b,c^	7.6	5.2	11.6	1.7	281.1	273.1
	700 g/t	43.4 ^a,b^	371.9 ^b,c^	178.5 ^c^	7.6	4.7	10.6	1.4	254.3	267.7
	1000 g/t	44.3 ^a^	374.9 ^a,b,c^	179.2 ^b,c^	7.5	3.9	9.6	1.4	197.1	263.2
SEM	-	1.3	2.7	1.2	0.1	0.5	0.6	0.5	22.1	11.2
*p*-Value Energy Level × Multi-Enzyme Dose	-	0.036	0.017	0.040	0.588	0.211	0.232	0.531	0.375	0.528
Main Effect										
Energy Level	STD	41.5	374.2	181.3	7.6	4.6	10.4	1.7	217.6	278.3
	STD-150 kcal/kg	42.3	375.1	180.7	7.5	4.8	10.8	1.7	234.4	265.4
	STD-200 kcal/kg	43.1	373.5	180.1	7.7	4.7	10.6	1.5	244.3	265.7
SEM	-	0.6	1.8	0.6	0.07	0.4	0.4	0.5	11.1	5.6
*p*-Value Energy Level	-	0.233	0.606	0.357	0.333	0.613	0.495	0.504	0.233	0.191
Linear	-	0.101	0.854	0.176	-	-	-	-	-	-
Quadratic	-	0.641	0.328	0.640	-	-	-	-	-	-
Main Effect										
Multi-Enzyme Dose	0	42.1	374.2	181.9	7.8	4.9	11.0	1.7	241.9	267.6
	350 g/t	42.2	373.4	180.3	7.6	4.8	10.8	1.6	242.8	271.4
	700 g/t	43.0	376.1	180.9	7.5	4.5	10.3	1.4	230.8	275.5
	1000 g/t	41.8	373.4	180.0	7.5	4.6	10.4	1.8	212.8	264.8
SEM	-	0.7	1.9	0.7	0.08	0.4	0.4	0.5	12.8	6.5
*p*-Value Multi-Enzyme Dose	-	0.731	0.416	0.173	0.079	0.613	0.313	0.400	0.320	0.676
Linear	-	0.944	0.924	0.063	-	-	-	-	-	-
Quadratic	-	0.410	0.494	0.759	-	-	-	-	-	-

^1^ Values with different superscripts (a, b, c) differ (*p* ≤ 0.05).

## Data Availability

The data presented in this study are available on request from the corresponding author. The data are not publicly available due to privacy agreement with the external funding body.

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
