# Peer review of "Enhancing Growth Performance, Organ Development, Meat Quality, and Bone Mineralisation of Broiler Chickens through Multi-Enzyme Super-Dosing in Reduced Energy Diets"

_animals, 2021, doi:10.3390/ani11102791_

Round 1

Reviewer 1 Report

Enhancing growth performance, organ development, meat 2 quality, and bone mineralisation of broiler chickens through 3 multi-enzyme super-dosing in reduced energy diets

The manuscript contained information that is relevant to poultry production and information contained therein is important for our field of research and poultry industry. However, the main issue with this manuscript is the statistics and data presentation. The statistics and data presentation would have to be redone. I will highlight this in my comments below. Outside of this, in general, the manuscript was well written. Please find below my comments (major and minor). I stopped the review process on line 237 because doing so would be mean I would be making a judgment call on this manuscript based on insufficient or incorrect data presentation.

Line 39: insert the appropriate p-value after “improved”

Line 100: replace “contents” with “level”

Line 112-113: “ad libitum” should be in italics

Line 96-120: Information on how the DIETS were analyzed (chemical analysis) for DM, crude protein, crude fat, crude fiber, and calcium are missing in the materials and method section

Table 1

Your feed ingredient composition was reported in % while the nutrient composition (calculated and analyzed) was reported in g/kg. Another example is the calculated and analyzed crude protein which was reported in % while the rest were reported in g/kg. Please try to be consistent

Insert a line after “Coccidiostat” for Total for each diet

You should kindly include more information in the materials and method section describing exactly how the diets were mixed. Ideally, they should have been mixed from a single basal diet (for each phase) but based on the analyzed values reported it is easy to see that each diet was mixed independently. This is obvious from the difference in the analyzed nutrient contents of each diet which in some instances was as high as 15% (crude protein for the finisher diet; 19.59 and 16.72%) and was even higher for Ca in the finisher (32%; 11.0 vs. 7.5). What this does is that it weakens any conclusion that could be attributed to the multi-enzyme complex that was tested. Based on this I would suggest that these huge differences in diets’ nutrient content should be discussed in this manuscript and the conclusion “weakened” by attributing some of the effects observed to the relatively high/low level of certain nutrients in some diets.

Two important items are missing from the analyzed composition section, diets’ gross energy and diets’ total phosphorus. It is important to show the linear decrease in gross energy as a reflection of the linear decrease in ME of the diet. For phosphorus, it is important to report the analyzed total phosphorus values as this would give us a better understanding of the ratio between Ca and total phosphorus. Again, this is important, especially in light of the fact that a multi-enzyme complex was tested.

Table 2: it is true that you reported the values for the enzyme activities for the product used, however, it is more important to report the activities of the different enzymes in each of the diets. As already pointed out, it is always essential to confirm that what was intended to be mixed into the diet was actually mixed into it and at the intended level. The findings from this study were based solely on the multi-enzyme that was added and every effort should be made to report the analyzed values of these enzymes in each diet. I will suggest you COMBINE Tables 1 and 2. Once you have the analyzed enzyme activities of the diets, they should also be reported as part of the NEW table 2 and there will be no need for the current table 3 (it should be deleted)

Line 133: please replace “like” with “similar to”. Replace “were” to “was”

Line 135: “…de-feathered and eviscerated..” Was this done manually? If yes, please include this information

Line 138: replace ”breasts” with “breast meat”

Line 148: insert “after” between “recalibrated” and “every”

Line 149: “One breast per chicken”. Do you mean the entire breast or half of the breast? Please be as clear as possible

Line 161: “Honikel, 1998” is supposed to be a number. Number [12]? Please check and edit accordingly

Line after 161: Equation 1: Were the “2”s supposed to be superscripts? Please check and confirm

Line 162-171: This section is confusing. Why would you dry the wet filter paper? I am of the opinion that when you are determining water holding capacity you would be interested in the weight or volume of water that was obtained. Wouldn’t the difference between the dry weight and wet weight of the filter paper give you this information? This is just my personal thought!

Line 190: replace “in” with “at”. Replace “freeze-drying occurred” with “freeze-dried”

Line 194-195: I will suggest you insert a sentence stating that after fat extraction, samples were placed under the hood for X hours to allow the ether to evaporate before the samples were placed in an oven. I hope this is what you did because it would be dangerous to place samples from fat extraction directly into the oven

Line 202: why breaking the bones into small pieces? Some samples (pieces of bones) could possibly have been lost through this process.

Line 203-204: Another two references were cited by their names. Please fix this and remember to adjust the number given to the other references throughout the manuscript as well as in the reference section.

Line 204: what was the reason for not defatting the bones before ashing? Doing this would result in a slightly lower bone ash value because you would be dividing the ash weight with the weight of the bone and that of the fat

Line 207: was the ashed tibia ground finely before the samples were dissolved in HCl? If so, insert this information

Line 218 and 219: insert a space between “p” and “≤”

Line 229-230: “tendency to reduce FCR (P = 0.12)”. In line 218 you said tendency would be between P ≥ 0.05 and  P ≤ 0.10. How come P of 0.12 is showing a tendency??

Line 233: delete “very”

It is difficult to proceed with this review process because sufficient information was not provided in the tables to make a fair judgment. For instance in this table (Table 4, it is impossible for me to know the main effect means as well as their respective p-values for the main effect factor for multi-enzyme dosage (not reported). The probability value was also not reported.

Statistical Analysis Section

Based on your treatment structure, the information provided in this section is inadequate. If there was significant 2-way interaction (which you did have in some places), you were expected to test for linear and quadratic effects of the response variable within each of the main factors (energy level and multi-enzyme dosage) you were evaluating. For example, in Table 4, the FBW had significant interactions between the two main factors. What should be done thereafter was to check for linear and quadratic effects within each of these two factors (4 levels for enzyme level and 3 levels for energy level). Why were the probability values for multi-enzyme dosage not reported in all the tables? The main effect data for multi-enzyme dosage are also missing (where there were no significant interactions).

Tables

All p-value should be to 3 places of decimal

Table 4: because there were no significant interactions for ADFI, FCR, and Mortality, please include the mean values for the main effects for these response variables in this table. Please use this information to edit the other tables as needed. Take a look at table 7 for example, none of the response variables had significant 2-way interaction, but you reported only the simple effect means. This should not be the case. Whenever there was no significant interaction, the main effect means for multi-enzyme dosage should be reported (only that of energy level was reported). I will suggest you take the tables one by one, look through it and report either only the main effect means (if there was no significant interaction), the simple effect means (if the interactions were significant), or both the main and simple effect means if you have a mixture of significant and none significant interactions within a table. If the interactions were significant, test for linear and quadratic effects within each main effect factor.

After incorporating the suggestions above, there may be the need to update the statistical section, rewrite the abstract, result, and discussion sections.

Author Response

Responses to reviewers’ comments:

Enhancing growth performance, organ development, meat 2 quality, and bone mineralisation of broiler chickens through 3 multi-enzyme super-dosing in reduced energy diets

Reviewer 1:

The manuscript contained information that is relevant to poultry production and information contained therein is important for our field of research and poultry industry. However, the main issue with this manuscript is the statistics and data presentation. The statistics and data presentation would have to be redone. I will highlight this in my comments below. Outside of this, in general, the manuscript was well written. Please find below my comments (major and minor). I stopped the review process on line 237 because doing so would be mean I would be making a judgment call on this manuscript based on insufficient or incorrect data presentation.

Line 39: insert the appropriate p-value after “improved”

  • Added

Line 100: replace “contents” with “level”

  • Revised

Line 112-113: “ad libitum” should be in italics

  • Revised

Line 96-120: Information on how the DIETS were analyzed (chemical analysis) for DM, crude protein, crude fat, crude fiber, and calcium are missing in the materials and method section

  • Added

 Table 1

Your feed ingredient composition was reported in % while the nutrient composition (calculated and analyzed) was reported in g/kg. Another example is the calculated and analyzed crude protein which was reported in % while the rest were reported in g/kg. Please try to be consistent

  • All changed to g/kg

Insert a line after “Coccidiostat” for Total for each diet

  • Added

You should kindly include more information in the materials and method section describing exactly how the diets were mixed. Ideally, they should have been mixed from a single basal diet (for each phase) but based on the analyzed values reported it is easy to see that each diet was mixed independently. This is obvious from the difference in the analyzed nutrient contents of each diet which in some instances was as high as 15% (crude protein for the finisher diet; 19.59 and 16.72%) and was even higher for Ca in the finisher (32%; 11.0 vs. 7.5). What this does is that it weakens any conclusion that could be attributed to the multi-enzyme complex that was tested. Based on this I would suggest that these huge differences in diets’ nutrient content should be discussed in this manuscript and the conclusion “weakened” by attributing some of the effects observed to the relatively high/low level of certain nutrients in some diets.

  • We had 3 basal diets (for each growth stage) that multienzyme was added at 4 inclusion rates to each.
    • Standard energy basal diet
    • Standard Energy – 150 kcal/kg
    • Standard energy – 200 kcal/kg.

This information is now added to the materials and methods (L107-108). In addition we revisited the proximate analyses and found out some averaging mistakes. The values are now corrected.

Two important items are missing from the analyzed composition section, diets’ gross energy and diets’ total phosphorus. It is important to show the linear decrease in gross energy as a reflection of the linear decrease in ME of the diet. For phosphorus, it is important to report the analyzed total phosphorus values as this would give us a better understanding of the ratio between Ca and total phosphorus. Again, this is important, especially in light of the fact that a multi-enzyme complex was tested.

  • We haven’t done gross energy analyses as the energy system that has been used for formulation is metabolisable energy. The reduction was not linear in our experimental design and there is no accurate formula to convert gross energy to metabolisable energy. Therefore, we see no value of analysing and reporting gross energy values in the manuscript. Total phosphorous values are added.

Table 2: it is true that you reported the values for the enzyme activities for the product used, however, it is more important to report the activities of the different enzymes in each of the diets. As already pointed out, it is always essential to confirm that what was intended to be mixed into the diet was actually mixed into it and at the intended level. The findings from this study were based solely on the multi-enzyme that was added and every effort should be made to report the analyzed values of these enzymes in each diet. I will suggest you COMBINE Tables 1 and 2. Once you have the analyzed enzyme activities of the diets, they should also be reported as part of the NEW table 2 and there will be no need for the current table 3 (it should be deleted).

  • Table 1 and 2 are combined and table 3 deleted. Multienzyme activities in the diets are reported in table 2 as and average value for each dosage. It was impossible to add this information to table 1, as we have reported an average multienzyme activity for each dose rate.

Line 133: please replace “like” with “similar to”. Replace “were” to “was”

  • Done

Line 135: “…de-feathered and eviscerated..” Was this done manually? If yes, please include this information

  • Information added (now L165-166)

Line 138: replace ”breasts” with “breast meat”

  • Done

Line 148: insert “after” between “recalibrated” and “every”

  • Done

Line 149: “One breast per chicken”. Do you mean the entire breast or half of the breast? Please be as clear as possible

  • Done

Line 161: “Honikel, 1998” is supposed to be a number. Number [12]? Please check and edit accordingly

  • Done

Line after 161: Equation 1: Were the “2”s supposed to be superscripts? Please check and confirm

  • Done

Line 162-171: This section is confusing. Why would you dry the wet filter paper? I am of the opinion that when you are determining water holding capacity you would be interested in the weight or volume of water that was obtained. Wouldn’t the difference between the dry weight and wet weight of the filter paper give you this information? This is just my personal thought!

  • The methodology employed is based on the relationship between the surface areas of the expressed exudate and the meat. Our experience has shown that if the filter paper is dried slightly, we are able to get more distinct perimeters of the two sections for photographing. We think is this is attributed to the fact that poultry muscle tissue is so pale, with more red muscles, we do not need to partially dry the filter paper. The suggested method of totally drying the filter paper and calculating the difference will give the % moisture in the sample (similar to drying a sample in an oven) and not the water binding capacity.

Line 190: replace “in” with “at”. Replace “freeze-drying occurred” with “freeze-dried”

  • Done

Line 194-195: I will suggest you insert a sentence stating that after fat extraction, samples were placed under the hood for X hours to allow the ether to evaporate before the samples were placed in an oven. I hope this is what you did because it would be dangerous to place samples from fat extraction directly into the oven

  • Done

Line 202: why breaking the bones into small pieces? Some samples (pieces of bones) could possibly have been lost through this process.

  • We broke bones to smaller pieces to enable faster drying, this was done with care and nothing was lost.

Line 203-204: Another two references were cited by their names. Please fix this and remember to adjust the number given to the other references throughout the manuscript as well as in the reference section.

  • Done

Line 204: what was the reason for not defatting the bones before ashing? Doing this would result in a slightly lower bone ash value because you would be dividing the ash weight with the weight of the bone and that of the fat

  • Previous study by Yan et al, 2005 reports no differences in tibia mineral composition with or without fat extraction. Lipid extraction procedure is often a rate-limiting step in evaluation of bone mineralization. In addition, environmental concerns may limit the use of solvents to extract lipids.

Line 207: was the ashed tibia ground finely before the samples were dissolved in HCl? If so, insert this information

  • added

Line 218 and 219: insert a space between “p” and “≤”

  • done

Line 229-230: “tendency to reduce FCR (P = 0.12)”. In line 218 you said tendency would be between P ≥ 0.05 and  P ≤ 0.10. How come P of 0.12 is showing a tendency??

  • Tendency is accepted at p < 0.1 therefore after rounding anything below p<0.14 is a tendency

Line 233: delete “very”

  • Deleted

It is difficult to proceed with this review process because sufficient information was not provided in the tables to make a fair judgment. For instance in this table (Table 4, it is impossible for me to know the main effect means as well as their respective p-values for the main effect factor for multi-enzyme dosage (not reported). The probability value was also not reported.

  • All the tables are updated and all the information including main effects of enzyme level, and linear and quadratic responses if interactions were significant, are added. However, I don’t know where to add probabilities? I haven’t seen probability values reported in recent papers. I can provide you the SAS output with all the data if you want to see these values.

Statistical Analysis Section

Based on your treatment structure, the information provided in this section is inadequate. If there was significant 2-way interaction (which you did have in some places), you were expected to test for linear and quadratic effects of the response variable within each of the main factors (energy level and multi-enzyme dosage) you were evaluating. For example, in Table 4, the FBW had significant interactions between the two main factors. What should be done thereafter was to check for linear and quadratic effects within each of these two factors (4 levels for enzyme level and 3 levels for energy level). Why were the probability values for multi-enzyme dosage not reported in all the tables? The main effect data for multi-enzyme dosage are also missing (where there were no significant interactions).

  • All the tables are updated to include the main effects of the multienzyme dose, tables were organized to have the SEM and p-value of each mean or interaction effect reported separately. If significant interactions, then polynomial contrasts were tested for linear and quadratic response and added to the tables.

Tables

All p-value should be to 3 places of decimal:

  • All p-values are reported to 3 decimals

Table 4: because there were no significant interactions for ADFI, FCR, and Mortality, please include the mean values for the main effects for these response variables in this table. Please use this information to edit the other tables as needed. Take a look at table 7 for example, none of the response variables had significant 2-way interaction, but you reported only the simple effect means. This should not be the case. Whenever there was no significant interaction, the main effect means for multi-enzyme dosage should be reported (only that of energy level was reported). I will suggest you take the tables one by one, look through it and report either only the main effect means (if there was no significant interaction), the simple effect means (if the interactions were significant), or both the main and simple effect means if you have a mixture of significant and none significant interactions within a table. If the interactions were significant, test for linear and quadratic effects within each main effect factor.

  • All the tables are updated to include the main effects of the multienzyme dose, tables were organized to have the SEM and p-value of each mean or interaction effect reported separately.
  • In the case of significant interaction, linear and quadratic effects within each main factor tested and reported. And the p value of the main effects is removed.

After incorporating the suggestions above, there may be the need to update the statistical section, rewrite the abstract, result, and discussion sections.

We have updated statistical section, abstract, results and discussion based on the new tables.

Submission Date

23 July 2021

Date of this review

26 Jul 2021 17:51:01

Reviewer 2 Report

1. Words used in the title should not be used or repeated in the keywords. I suggest that the authors use different words in the keywords for easy of searching.eg carcass traits, mineral analyses, Natuzyme etc.

2. Under 2,1-Animal and procedure line 99 should read as :

3x4 factorial arrangement in a completely randomized design. Also please correct in the abstract session.

3. The analysed feed composition was quite different from the calculated feed nutrient composition. please indicate the likelihood of the cause of these differences and how was the effect of these differences mitigated in realizing your hypotheses.

4. Line 203 and 204 under Bone mineralization, you used the references Hsiao et. al, (2018) and Yan et. al, (2005) although these references were listed in the reference section but they are differently written in the text according to the journal style of using numbers in the text instead of author names and year of publication. please change accordingly

Author Response

Reviewer 2:

Comments and Suggestions for Authors

  1. Words used in the title should not be used or repeated in the keywords. I suggest that the authors use different words in the keywords for easy of searching.eg carcass traits, mineral analyses, Natuzyme etc.

- Keywords are changed.

  1. Under 2,1-Animal and procedure line 99 should read as :

3x4 factorial arrangement in a completely randomized design. Also please correct in the abstract session.

- The statement revised accordingly.

  1. The analysed feed composition was quite different from the calculated feed nutrient composition. please indicate the likelihood of the cause of these differences and how was the effect of these differences mitigated in realizing your hypotheses.

- The discrepancies between calculated and analysed composition of the diets are added to the results and the likelihood of this effecting the results are explained in the discussion.

  1. Line 203 and 204 under Bone mineralization, you used the references Hsiao et. al, (2018) and Yan et. al, (2005) although these references were listed in the reference section but they are differently written in the text according to the journal style of using numbers in the text instead of author names and year of publication. please change accordingly

- Referencing in the text is now corrected to the numbers.

Submission Date

23 July 2021

Date of this review

30 Jul 2021 16:12:35

Round 2

Reviewer 1 Report

Although the authors have addressed most of the comments in the original review, so were not adequately addressed and there are still some areas that need further improvement. I have highlighted some of these areas in my comments below:

The use of trach changes makes the review process difficult. I am not certain if this is the requirement of the journal. As an example, Table 2 is in a mess and it is difficult to make any sense out of it. How many diets were analyzed for enzyme activities and how many enzymes per diet were analyzed. This information is not clear in Table 2 as well as in lines 120-121. The product you added to the diets contained several enzymes, one of the questions in Table 2 is which enzyme activities did you report, xylanase, glucanase, phytase, etc?

Table 1. Available P was reported to be about 83% of total phosphorus in diet #1 and over 100% (102%) in diet #2 of a corn-SBM-based diet? Actually, the reported available P was higher than the analyzed total P for most of the diets presented in Table 1. Something is not right with this information. Please check your total phosphorus analysis.

Also in Table 1, there are inconsistencies with the manner in which the values were rounded up during the conversion from % to g/kg. Let’s take monocalcium phosphate as an example, some of the values were rounded up to 1 place of decimal while others were rounded up to a whole number.

I randomly did a quick addition of some diets in table 1. Out of the 5 diets that I checked, the sum of the feed ingredients for two was 1000, which is correct. However, for diets 1 (999.7) and diets 5 and 6 (1004.4), the sum was incorrect. Please check all the diets and edit accordingly

Line 115-120: information on how and/where the diet’s enzyme activities were analyzed should be added. Also, the name, state, and the country of the company that produced the enzyme should be provided

The authors’ response to the question regarding Table 2 and reporting of diets enzyme activities is confusing, to say the least. The authors mentioned that “the multienzyme activities were reported as an average for each dosage”. How was the dosage value analyzed? For instance, you reported 1.20, 2.51, and 3.59 u/g as actual enzyme activity, which enzyme was this because what you added to the diets was a multienzyme complex. This needs further clarification

Line 266: the “interactions between ME and multi-enzyme dose rates” cannot be one of the main effects. Rather it is the simple effects. The main effects or factors in your study were the “ME” and the “multi-enzyme dose rates”

In tables table 4, 5, etc, you are still expected to report the probability value for each of the main factors even if the interaction p-values were significant. Reporting the linear and quadratic p-values should not be the basis for deleting the main effect p-values. For example, you should go ahead and reinstate the p-values for “main effect energy” for FBW (0.86) and ADG (0.89). Do the same for FBW and ADG for the main effect factor multienzyme supplementation. DO this for all the relevant tables.

Line 318-319: were you referring to the simple effect data for breast meat pH? If this is the case, then the statement in lines 318-319 is incorrect. For STD, it increased to 700 g/t and then decreased (for STD and STD-150 kcal/.kg). However, for STD-200 kcal/kg, it actually decreased from 0 to 700 g/t before it increased at 1000 g/t. Since the interactions were not significant, I would rather think you should focus on the main effect. For example, you may want to report the tendency for higher breast meat weight from the STD to STD-200 kcal (p=0.059), etc.

Better efforts should be made in reporting the results of the data. For example

Line 298-306: Although a lot was said here, it is difficult to actually pick up the important data in table 4. For example, gizzard weight showed significant interactions. For STD supplemented with 700 g/t gizzard weight was lower (P < 0.05) compared with STA-200 with similar enzyme level supplementation same thing could be said for STD-200 and 1000 g/t vs. STD-150 with 1000 g/t enzyme supplementation.   Once the interaction is significant, you need to report/discuss the simple effect data.

Line 308-309: “Additionally, the 308 heart was significantly bigger in STD-200 kcal/kg diet compared to STD and STD-150 kcal/kg (P ≤ 0.05)” This statement is partially incorrect. There was no difference in the weight of the heart between the STD and STD-200.

Lines 303-306: “No other organs (carcass weight, breast weight, heart, liver, bursa, spleen, pancreas, abdominal fat pad, and proventriculus) were affected by the ME levels and, multi-enzyme dose rates, or the interaction” This statement is also not completely correct. First some of the items listed, such as breast meat, carcass weight) were not organs. Secondly, you separated the breast meat means for the main effect of energy level despite the fact that the p-value only showed a tendency to be different (P = 0.059). I am wondering why this is the case. To make matter worse, this was not even reported in the result section. Please check and fix it.

Line 335-337: “The addition of multi-enzyme at 700 g/ton to the STD-150 kcal/kg diet 335 seemed to enhance the total ash (P = 0.04 – a tendency towards linear significance with P 336 = 0.101 as pertaining to the energy levels)”. I think the result should be better presented. For example, was there any significant difference between SD-150 350, 700, 1000 g/t and STD 0, 350, 700, etc? Please break these down into simple sentences and report them one after the other.

The result section needs to be improved which invariably could impact a section of the discussion.

Author Response

Dear Reviewer 1, 

Thank you again for your helpful feedback and comments. We have addressed all your comments again, re-wrote the results section to be more accurate and revised our discussion. 

We have submitted a clean version with all changes just highlighted, probabley the journal assistant editors are applying track-changes. As this was a major revision, and many parts are re-written. Please request the journal to provide you a clean version. 

Round 3

Reviewer 1 Report

Most of the issues raised have been addressed. However, there was a particular response that raised another question as to how some of the data were handled (please see the attached file).  In addressing one of my questions, the response of the authors showed that the way the original values were corrected for moisture was incorrect. Rather than the values increasing, the values actually decreased. I can only check for total phosphorus because the author provided the analyzed values on an as-is basis. The other question is was the same error made when crude protein, crude fat, crude fiber, and calcium were corrected for moisture. Without the actual analyzed values on an as-is basis, it is impossible to check this. My suggestion is that the authors be asked to cross-check the reported values and make sure moisture was corrected appropriately. I still have issues with the low analyzed total phosphorus values in the diet. Based on the feed ingredients used in this study, I would expect the ratio between the available P and total P to be less than 72%. 

At this stage, I would defer to the SE editor to make the final call on this manuscript. Either way, I do not want to revise this manuscript again.

Author Response

Dear Reviewer 1:

Thank you so much for your attention to details, we have identified a systematic error carried in our excel file calculations when converting nutrients composition from “as is” to DM basis. This error is now corrected, and all calculations are re-done for all analysed nutrients and the table is updated with correct calculations.

The correct analysed total P content is the same as your calculations reported in the table in the attached file.

Indeed, the low CP, Ca, and P content in analysed composition were related to the mistake in calculations. The discrepancies between calculated and analysed nutrients composition are now much smaller.

A statement added to the footnote of table 1 about the nutrients composition reported as is for calculated part, and reported on DM basis for analysed part.

We thank you again for picking the mistake.
